# Mitochondrial ubiquitin ligase alleviates Alzheimer's disease pathology via blocking the toxic amyloid-β oligomer generation

Keisuke Takeda[1,5,6], Aoi Uda[1], Mikihiro Mitsubori[1], Shun Nagashima[1], Hiroko Iwasaki[1], Naoki Ito[1,2], Isshin Shiiba[1,2], Satoshi Ishido [3], Masaaki Matsuoka[4], Ryoko Inatome[1,2] & Shigeru Yanagi [1,2 ✉]

Mitochondrial pathophysiology is implicated in the development of Alzheimer's disease (AD). An integrative database of gene dysregulation suggests that the mitochondrial ubiquitin ligase MITOL/MARCH5, a fine-tuner of mitochondrial dynamics and functions, is down-regulated in patients with AD. Here, we report that the perturbation of mitochondrial dynamics by MITOL deletion triggers mitochondrial impairments and exacerbates cognitive decline in a mouse model with AD-related Aβ pathology. Notably, MITOL deletion in the brain enhanced the seeding effect of Aβ fibrils, but not the spontaneous formation of Aβ fibrils and plaques, leading to excessive secondary generation of toxic and dispersible Aβ oligomers. Consistent with this, MITOL-deficient mice with Aβ etiology exhibited worsening cognitive decline depending on Aβ oligomers rather than Aβ plaques themselves. Our findings suggest that alteration in mitochondrial morphology might be a key factor in AD due to directing the production of Aβ form, oligomers or plaques, responsible for disease development.

[1] Laboratory of Molecular Biochemistry, School of Life Sciences, Tokyo University of Pharmacy and Life Sciences, Hachioji, Tokyo, Japan. [2] Laboratory of Molecular Biochemistry, Department of Life Science, Faculty of Science, Gakushuin University, Toshima-ku, Tokyo, Japan. [3] Department of Microbiology, Hyogo College of Medicine, Nishinomiya, Japan. [4] Department of Pharmacology, Tokyo Medical University, Shinjuku-ku, Tokyo, Japan. [5] Present address: Department of Biology, University of Padova, Padova, Italy. [6] Present address: Venetian Institute of Molecular Medicine, Padova, Italy. ✉email: syanagi@ls.toyaku.ac.jp

Alzheimer's disease (AD) is the most prevalent neurodegenerative disease, which is characterized by memory and cognitive impairments. The aggregation of free amyloid-beta (Aβ) monomers in the brain is an initial step in the development of AD. Therefore, plaques consisting of aggregates of Aβ, such as its fibrils and prefibrils, are a major hallmark of AD. On the other hand, increasing reports have suggested that, in addition to Aβ plaques, Aβ oligomers are also pathogenic forms of Aβ[1–3]. In fact, a familial AD mutation within the Aβ sequence (E22Δ) has been shown to lead to the dominant accumulation of oligomeric Aβ without subsequent plaque formation, causing the early onset of AD[4]. Likewise, in several patients with sporadic AD, a high correlation between the clinical stage and the amount of soluble Aβ oligomers, rather than that of insoluble Aβ including fibrils and plaques, was identified[5,6]. These findings show that soluble Aβ oligomers are accumulated in AD cases as off-pathway oligomers unrelated (or less related) to the further polymerization into larger units such as fibrils and plaques. However, the in vivo mechanism directing the generation of the Aβ form predominantly responsible for Aβ pathology, namely, oligomers or plaques, remains largely unknown. Aβ oligomers potentially cause severe neurological injury compared with Aβ fibrils or plaques themselves[7,8], implying that undirected conversion from Aβ oligomers to plaques reduces the neurotoxicity of Aβ. Therefore, the dominant accumulation of toxic Aβ oligomers might be a key factor in Aβ pathology.

Mitochondrial malfunctions are common hallmarks of AD, suggesting that mitochondrial homeostasis plays a pivotal role in AD development[9,10]. To avoid defects in mitochondrial functions, individual mitochondria are habitually integrated by dynamic structural remodeling via fusion and fission events. Mitochondrial fusion allows functional complementation by mixing the materials between damaged and healthy mitochondria[11,12]. Asymmetric mitochondrial fission eliminates abnormal contents from individual mitochondria[11,12]; hence, an organized mitochondrial network is a physiological foundation for maintaining mitochondrial quality. It is well reported that several pathological conditions of AD, including Aβ pathology, disturb mitochondrial dynamics through direct or indirect processes[13]. Therefore, the disruption of mitochondrial dynamics may be an intrinsic etiological factor primarily initiating, or at least aggravating, the mitochondrial pathophysiology in AD. In addition, it is still unclear how the pathogenesis by Aβ is altered by the perturbation of mitochondrial dynamics and the subsequent mitochondrial pathophysiology.

Previously, we identified the mitochondrial E3 ubiquitin ligase (MITOL/MARCH5), an integral mitochondrial outer membrane protein[14]. We and other research groups have demonstrated that MITOL serves as a regulator for maintaining optimal mitochondrial morphology, and also ER tethering, through comprehensive modulation by ubiquitinating substrates related to mitochondrial fusion or fission processes[14–18]. Recently, we have generated neuron-specific MITOL-knockout (KO) mice and reported that MITOL deletion in vivo causes the morphological abnormalities of mitochondrial structure[19]. An integrative database for gene dysregulation in AD, named Alzbase, has suggested that the expression level of the *mitol/march5* gene is significantly reduced in patients with AD[20]. Taking these findings together, MITOL downregulation might trigger or aggravate mitochondrial pathophysiology in AD by disorganizing the formation of mitochondrial networks.

In this paper, we report that MITOL deletion in a model mouse with AD-related Aβ pathology accelerated mitochondrial disconnection, followed by mitochondrial impairments, in the brain. Importantly, MITOL deletion enhanced the seeding activity of Aβ fibrils, but not the spontaneous formation of fibrillized plaques, inducing the excessive generation of toxic off-pathway Aβ oligomers from surrounding free Aβ monomers. Our findings may lead to the development of AD therapies targeting Aβ oligomers.

## Results

**MITOL is transcriptionally downregulated by Aβ**. APPswe/PSEN1dE9 transgenic mice referred to here as APP/PS1 mice are widely recognized as a mouse model for AD-related Aβ pathology. APP/PS1 mice contain the human transgene with the Swedish mutation (KM595/596NL) of APP (APPswe) combined with a deletion mutation of exon 9 in PS1 (PS1ΔE9). These mutations in APP and PS1 induce toxic Aβ production due to the destabilization of PS1 followed by γ-secretase abnormality. APP/PS1 mice exhibit Aβ plaques from 4 months of age and mild cognitive impairment from around 12–15 months of age by the AD-related Aβ pathology[21]. First, we investigated whether the gene expression of MITOL was downregulated in the model mouse with Aβ pathology, similar to the results of Alzbase[20]. As expected, both protein and mRNA levels of MITOL were reduced in the cerebral cortex of 15-month-old APP/PS1 mice compared with those in non-transgenic mice (Fig. 1a, b). We further used a cellular model that expresses APPswe combined with siRNA targeting PS1 (siPS1) instead of PS1ΔE9 (Supplementary Fig. 1a). Consistent with the results obtained from APP/PS1 brain, both protein and mRNA MITOL levels were decreased in cells co-expressing APPswe and siPS1 (Fig. 1c, d). The downregulation of MITOL by the co-expression of APPswe and siPS1 was recovered by treatment with the γ-secretase inhibitor DAPT, which blocks the generation of Aβ (Supplementary Fig. 1b, c). These results demonstrate that Aβ decreases MITOL expression, in a manner at least partly dependent on transcriptional regulation, in both mouse and cell models of Aβ pathology.

**MITOL deletion facilitates mitochondrial pathophysiology in the APP/PS1 brain**. To evaluate the causal relevance of MITOL downregulation to mitochondrial pathophysiology in Aβ pathology, cerebral cortex- and hippocampus-specific MITOL-deleted APP/PS1 mice were generated by crossing MITOL[F/F] mice with Emx1-Cre transgenic APP/PS1 mice (Supplementary Fig. 1d, e). We then analyzed individual mitochondrial morphology in vivo using transmission electron microscopy (TEM). In the APP/PS1 brain, MITOL deletion significantly increased the number of smaller mitochondria (Fig. 1e, f, Supplementary Fig. 1f). This indicates that the remaining expression of MITOL in the APP/PS1 brain (~40% compared with the level in non-transgenic mice) still has a sufficient protective effect on mitochondrial morphology. We also classified the internal structure of mitochondria into normal (Class I) or abnormal (Classes II and III), including dilated mitochondrial cristae and empty matrices (Fig. 1g). In addition to the reduction of mitochondrial size, mitochondria in the MITOL-deleted APP/PS1 brain exhibited disruption of the inner membrane compared with the observations in the other groups of mice (Fig. 1h). We then performed COX staining, which is a well-established method for monitoring mitochondrial respiratory activity in vivo. As expected, MITOL deletion accelerated alterations of mitochondrial bioenergetics in the APP/PS1 brain (Fig. 1i, Supplementary Fig. 1g). In addition, MITOL deletion reduced the mitochondrial activity for ATP production in the APP/PS1 brain (Fig. 1j). Consistent with this, MITOL knockdown in cells co-expressing APPswe and siPS1 decreased the level of oxygen consumption by mitochondria (Supplementary Fig. 1h–j). Taking these findings together, MITOL plays a protective role against mitochondrial dysfunction in Aβ pathology.

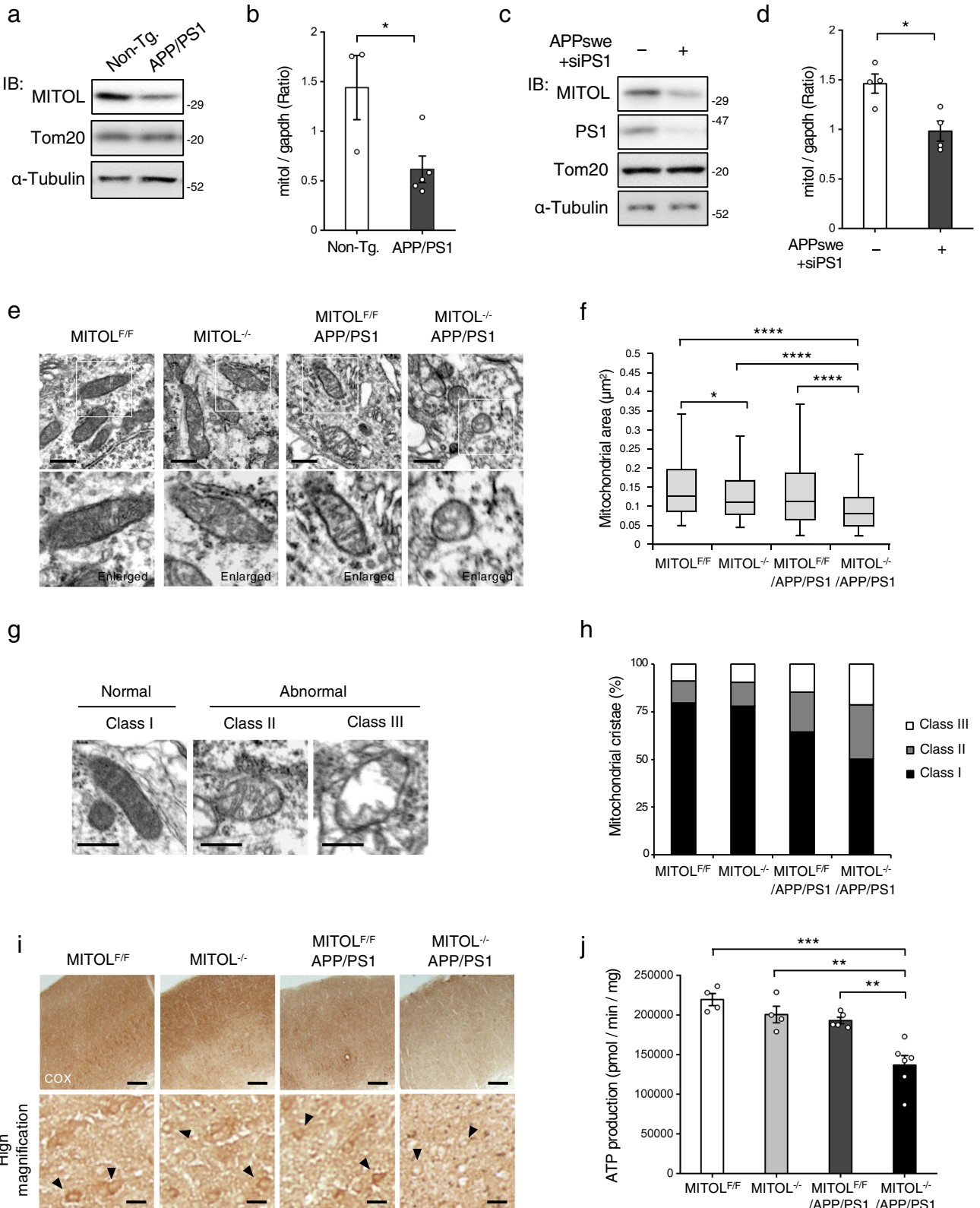

**MITOL deletion exacerbates brain impairments in APP/PS1 mice.** We examined the effects of MITOL deletion on behavioral deficits, such as memory and cognitive impairments, in Aβ pathology. The object recognition memory and spatial working memory were assessed by the novel object recognition test and Y-maze test, respectively, using 15-month-old mice (Fig. 2a, b). MITOL deletion itself did not cause any defects in these memory functions, whereas APP/PS1 mice developed mild memory impairments (Fig. 2a, b). MITOL-deleted APP/PS1 mice exhibited the most severe behavioral abnormalities among all groups of mice (Fig. 2a, b). Likewise, the Barnes maze test revealed that MITOL deletion led to worsening cognitive decline in APP/PS1 mice (Fig. 2c). No obvious difference in the movement activity of mice was observed in the open-field box or Y-maze apparatus

**Fig. 1 The combination of Aβ accumulation and MITOL loss enhances mitochondrial dysfunction. a–d** MITOL was downregulated by Aβ in the model mice and cells. Brain lysates were solubilized from the cerebral cortex of indicated mice at 15 months of age, followed by immunoblotting with indicated antibodies (**a**). Indicated mRNA levels in the cerebral cortex of indicated mice were measured by qRT-PCR (**b**). SH-SY5Y cells stably expressing APPswe were transfected with siPS1 48 h before each analysis (**c, d**). As control cells, SH-SY5Y without stable expression were transfected with scramble siRNA instead of siPS1. These cell lysates were immunoblotted with indicated antibodies (**c**) or analyzed by qRT-PCR (**d**). Error bars indicate ±SE (**b**: $n = 3–5$, **d**: $n = 4$). *$p < 0.05$ (Student's $t$ test). Non-Tg.: Non-transgenic MITOL$^{F/F}$ mice. APP/PS1: MITOL$^{F/F}$ APP/PS1 mice. **e–h** MITOL loss accelerated mitochondrial dysfunction in the model mice. The brain of indicated mice at 15 months of age was subjected to transmission electron microscopy (TEM) analysis (**e–h**) or COX staining (**i**). The lower panels show high-magnification images of the boxed regions (**e**). Arrowheads indicate neurons (**i**). The scatter dot plot indicates mitochondrial area. Horizontal bar, median; box limits, 25th and 75th percentiles; whiskers, 10th and 90th percentiles (**f**: $n =$ over 300 mitochondria in the cell body of neurons of the cerebral cortex). The morphology of mitochondrial cristae is classified as normal (Class I) or abnormal (Class II, III) (**g, h**). Mitochondria of indicated groups are distinguished into three types (**h**: $n =$ over 300 mitochondria in the cell body of neurons of the cerebral cortex). Scale bar represents 500 nm (**e, g**), 200 μm (**i**: upper panels), 20 μm (**i**: lower panels). In vitro mitochondrial ATP, production assay was performed as described in the method. Error bars indicate ±SE (**j**: $n = 4–6$). ***$p < 0.001$, **$p < 0.01$ (one-way ANOVA, Tukey's test).

(Supplementary Fig. 2a, b). These findings show that MITOL protects neurons against Aβ pathology. Aβ aggregates into oligomers and fibrils, which are generally accepted to trigger neuronal dysfunction and neuroinflammation in the brain of AD patients and Aβ pathology model mice, even before the onset of dementia symptoms. At 12 months of age, MITOL deletion decreased both the number of nissl-positive neurons and the size of soma in the cerebral cortex of APP/PS1 mice (Fig. 2d, e, Supplementary Fig. 2c). MITOL deletion in the hippocampus of APP/PS1 mice also led to significant reductions of the synaptic markers synaptophysin and PSD-95 proteins (Fig. 2f, g). In addition, MITOL-deleted APP/PS1 mice showed remarkably higher activation of microglia together with an increase of CD68, a marker of activated microglia, than the other groups of mice (Fig. 2h, Supplementary Fig. 2d, i). Moreover, the expression of pro-inflammatory cytokines TNFα, IL-1β, and IL-6 was upregulated by MITOL deletion in APP/PS1 mice (Supplementary Fig. 2e). These results support the notion that MITOL prevents Aβ-related pathogenesis including neuronal injury and neuroinflammation in vivo.

**MITOL deletion boosts the secondary aggregation of Aβ monomer-catalyzed Aβ fibrils without affecting Aβ metabolism.** Findings have suggested that Aβ impairs mitochondrial functions directly, accelerating cognitive decline and neuronal damage in AD[9,10]. Conversely, recent studies have also revealed the fact that mitochondria participate in the regulation of Aβ aggregation itself, although the mechanisms behind this remain incompletely understood[22–25]. The in vivo contribution of mitochondrial pathophysiology to Aβ aggregation also remains controversial[26–28]. Mitochondrial impairments, designed by different experimental approaches, lead to inconsistent outcomes that are represented as either increased or decreased Aβ plaque formation in AD-related Aβ pathology[26–28]. To understand how mitochondrial malfunctions are involved in the pathogenesis of Aβ, there is a clear need to reappraise the status of Aβ in Aβ pathology model mice combined with mitochondrial impairments. We, therefore, characterized the metabolism and aggregation of Aβ in detail, including its oligomeric state, using MITOL-deleted APP/PS1 brain.

The total amount of Aβ in the brain is determined by the balance between APP processing and Aβ clearance such as Aβ degradation. An imbalance of Aβ metabolism triggers the oligomerization and fibrillization of Aβ, which is considered to be an initial step in the development of AD. We first examined the in vivo effect of MITOL deletion on Aβ metabolism. The expression levels of *human app* mRNA in the cerebral cortex of APP/PS1 mice were not altered by MITOL deletion (Fig. 3a). In addition, APP processing products, full-length APP, α-CTF, and β-CTF, were not changed in the APP/PS1 brain and the

MITOL-deleted APP/PS1 brain (Fig. 3b, c). The hippocampus of MITOL-deleted APP/PS1 showed the gene expression of Aβ-degrading enzymes such as IDE, NEP, MMP2, and MMP9, at the same level as those of the APP/PS1 brain (Supplementary Fig. 3a). These observations led us to the idea that MITOL alleviates Aβ pathology independently of Aβ metabolism itself.

Aβ aggregation and plaque formation are major hallmarks of AD. To clarify the role of MITOL in Aβ plaque formation, we evaluated the Aβ plaques in APP/PS1 mice at 15 months of age by the combination of two staining methods for Aβ plaques[29–31]. Anti-Aβ antibody 6E10 potentially stains the overall region of each Aβ plaque[29–31]. Thioflavin S (ThS) is an indicator of fibril formation, thereby staining the fibrillized core (also known as the amyloid core) in each Aβ plaque[29–31]. Unexpectedly, MITOL deletion in the APP/PS1 brain affected neither the total number nor the overall size of toxic Aβ plaques with a fibrillized core, despite the exacerbated Aβ toxicity (Fig. 3d–f). These results suggest that toxic Aβ plaques were generated spontaneously at the same level regardless of MITOL expression. However, the Aβ plaques in the MITOL-deleted APP/PS1 brain expanded the fibrillized region (Fig. 3d, g). Likewise, MITOL deletion enhanced the area ratio of fibrils in each Aβ plaque (Fig. 3h). These findings imply that MITOL deletion facilitates the fibrillization only in or around Aβ plaques. Consistent with this, the formation of non-toxic Aβ plaques without any fibrillized core was severely reduced in MITOL-deleted mice (Fig. 3d, Supplementary Fig. 3b).

Recent reports indicated that Aβ fibrils, such as plaques with a fibrillized core, function as a seed for Aβ aggregation and thus catalyze the secondary assembly of free Aβ monomers more efficiently than the spontaneous self-assembly among free Aβ monomers[32,33]. These previous findings led us to the idea that Aβ fibrils in the MITOL-deleted APP/PS1 brain might be able to more efficiently catalyze the secondary assembly of free and dispersible Aβ monomers by increased seeding activity, thereby resulting in a drastic expansion of the fibrillized region in Aβ plaques. To investigate this possibility, we utilized a previously described method of purifying Aβ fibrils from the brain and detected the seeding activity (Supplementary Fig. 3c)[34,35]. Thioflavin T (ThT) and ThS confirmed the presence of highly concentrated fibrils after the purification (Supplementary Fig. 3d, e). The seeding effect of the purified and normalized Aβ fibrils was examined using ThT. Seed-free Aβ monomers showed a robust increase in the fluorescence of ThT after approximately 20 h of co-incubation with control solution by its spontaneous self-assembly (Fig. 3i, Supplementary Fig. 3f). The lag time up to the emergence of newly formed Aβ fibrils was shortened by the addition of Aβ fibrils isolated from the APP/PS1 brain (Fig. 3i, Supplementary Fig. 3f), indicating that the purified Aβ fibrils serve as aggregation seeds, similar to the findings in previous reports[34,35]. The equal amounts of Aβ fibrils isolated from the

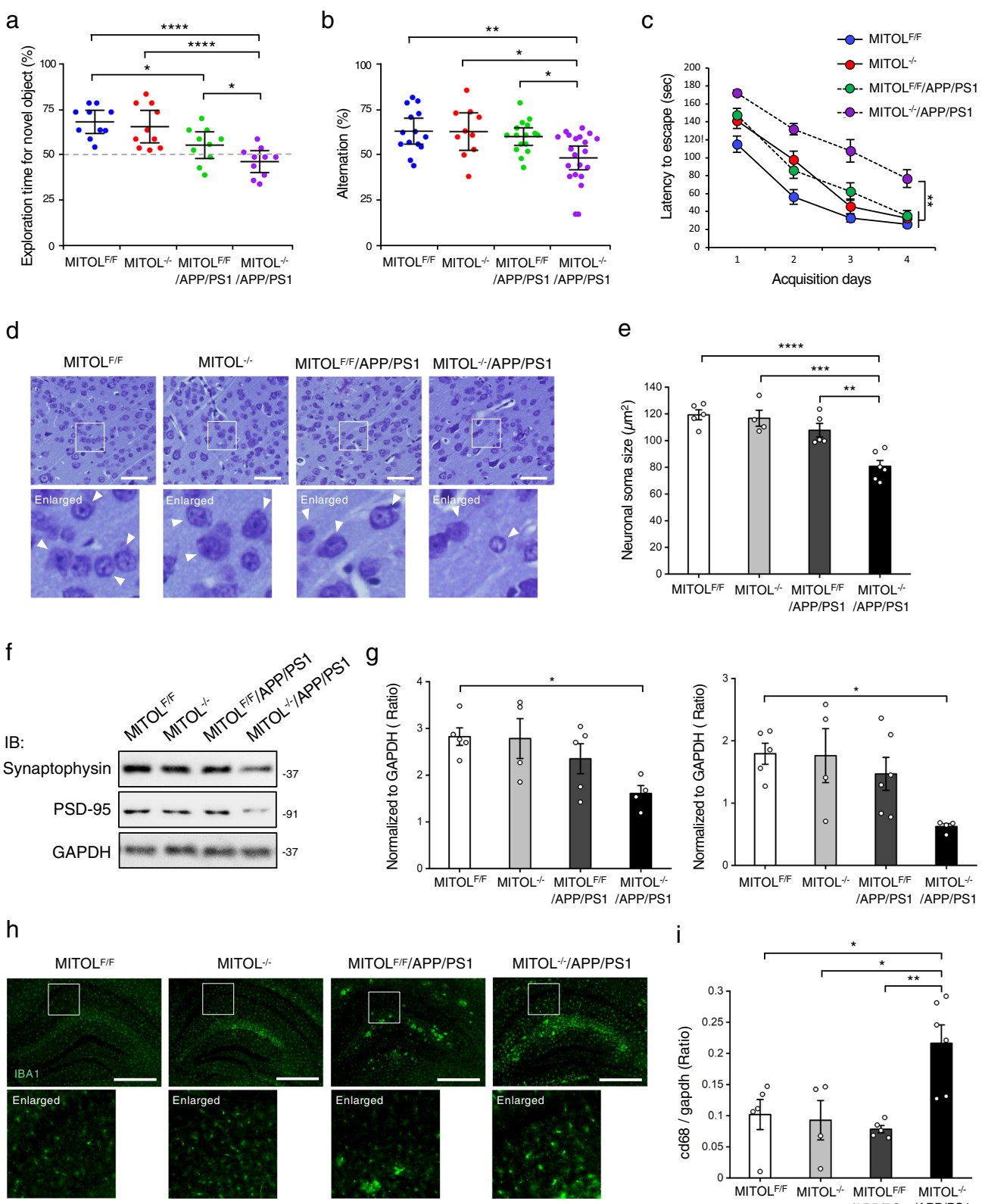

MITOL-deleted APP/PS1 brain displayed higher seeding activity than those isolated from the APP/PS1 brain (Fig. 3i, Supplementary Fig. 3f, g), mainly represented by the shortened lag time, thereby revealing that Aβ fibrils in the MITOL-deleted APP/PS1 brain are capable of initiating extension and maturation earlier than the same-sized Aβ fibrils in the APP/PS1 brain. This is consistent with the notion indicated in Fig. 3h that individual Aβ plaques in the MITOL-deleted APP/PS1 brain displayed an area highly occupied by a fibrillized core compared with those in the APP/PS1 brain. Interestingly, ELISA assays to detect types of Aβ species, such as Aβ38, Aβ40, Aβ42, and Aβ43, showed that Aβ43 specifically increased in the Aβ fibrils of the MITOL-deleted APP/PS1 brain compared with those in the APP/PS1 brain (Supplementary Fig. 3h). Collectively, these results suggest that

**Fig. 2 MITOL-deleted APP/PS1 mice show severe cognitive impairment, synapse alteration, and neuroinflammation. a–c** MITOL deletion exacerbated Aβ pathology-mediated cognitive decline. Indicated mice at 15 months of age were subjected to the novel object recognition test (**a**), Y-maze test (**b**), or Barnes maze test (**c**), respectively. Error bars indicate ±SE (**a**: $n = 10$, **b**: $n = 10$–20, **c**: $n = 10$–16). ****$p < 0.0001$, **$p < 0.01$, *$p < 0.05$ (one-way ANOVA, Tukey's test). **d**, **e** MITOL deletion enhanced neuronal atrophy in the AD model mice. Brain sections of indicated mice were stained using cresyl violet (**d**). The lower panels show high-magnification images of the boxed regions. Arrowheads indicate neurons. The average size of each neuronal soma stained with cresyl violet was calculated from over 300 cells using ImageJ (**e**). Scale bar represents 50 μm. Error bars indicate ±SE (**e**: $n = 4$–6). ****$p < 0.0001$, ***$p < 0.001$, **$p < 0.01$ (one-way ANOVA, Tukey's test). **f**, **g** Synapse markers, Synaptophysisn and PSD-95, were significantly reduced in the hippocampus of MITOL-deleted APP/PS1 mice at 12 months of age. Hippocampal lysates of indicated mice were immunoblotted with indicated antibodies. Error bars indicate ±SE (**g**: $n = 4$–6). *$p < 0.05$ (one-way ANOVA, Tukey's test). **h**, **i** Activated microglia was increased in MITOL-deleted APP/PS1 mice. Brain sections of indicated mice at 12 months of age were immunostained with anti-IBA1 antibody (**h**). The lower panels show high-magnification images of the boxed regions. Scale bar represents 100 μm. The indicated mRNA in the hippocampus extracted from indicated mice at 12 months of age was evaluated by qRT-PCR. Error bars indicate ±SE ($n = 4$–6). **$p < 0.01$, *$p < 0.05$ (one-way ANOVA, Tukey's test).

MITOL deletion altered not only the seeding activity but also the composition of Aβ fibrils, rather than their spontaneous formation, in vivo.

**MITOL deletion leads to excessive generation of Aβ oligomers by the higher seeding activity of Aβ fibrils, triggering severe cognitive symptoms.** Mounting evidence suggests that non-fibrillar soluble Aβ oligomers are another deleterious form of Aβ[1–3], although the relationship between Aβ oligomer accumulation and mitochondrial pathophysiology is poorly characterized. Because MITOL deletion did not alter the spontaneous formation of toxic Aβ plaques, despite the associated enhanced seeding activity (Fig. 3, Supplementary Fig. 3), we also investigated the accumulation level of another detrimental form of Aβ, namely, oligomers. Interestingly, the total levels of soluble Aβ were clearly elevated by MITOL deletion in APP/PS1 mice (Fig. 4a, b). Both the ELISA measurements and the dot blot analysis with Aβ oligomer-specific antibody revealed that MITOL deletion caused the aberrant accumulation of oligomeric Aβ in the APP/PS1 brain (Fig. 4c, Supplementary Fig. 4a, b). Interestingly, soluble Aβ was generated at the same level in the APP/PS1 and MITOL-deleted APP/PS1 brains at 3 months of age before extracellular Aβ plaque formation (Supplementary Fig. 4c–e), suggesting that the dominant generation of oligomeric Aβ in the MITOL-deleted APP/PS1 brain might be dependent on the formation of extracellular Aβ plaques, in other words, secondary assembly by the fibrillized form of Aβ plaques. Therefore, we analyzed the effect of seeding on the polymerization of free Aβ monomers using Bis-ANS, which is used to measure the solvent-exposed hydrophobicity of misfolded proteins and reported to strongly detect oligomeric assemblies of Aβ rather than Aβ fibrils[36,37]. As expected, Aβ monomers showed an efficient increase in the Bis-ANS fluorescence in the presence of Aβ fibrils isolated from the MITOL-deleted APP/PS1 brain, rather than the equal amount of Aβ fibrils isolated from the APP/PS1 brain (Fig. 4d). Importantly, Aβ fibrils derived from the MITOL-deleted APP/PS1 brain showed not only a shortened lag time but also the accelerated aggregation in secondary oligomerization (Supplementary Fig. 4f, g). Thus, Aβ fibrils in the MITOL-deleted APP/PS1 brain are capable of initiating secondary oligomerization earlier and additionally producing new oligomers faster than the same-sized Aβ fibrils in the APP/PS1 brain. The enhanced oligomerization of free Aβ monomers by Aβ fibrils derived from the MITOL-deleted APP/PS1 brain was also confirmed by the Aβ oligomer-specific antibody (Supplementary Fig. 4h, i). Taking these findings together, MITOL deletion robustly enhanced the generation of Aβ oligomers by Aβ fibril-catalyzed secondary assembly. Numerous studies have defined two distinct types of Aβ oligomers based on the immunoreactivity of conformational antibodies, named OC or A11 antibody[38,39]. OC-recognized Aβ oligomer has a parallel β-sheet structure similar to that of Aβ fibrils and thus acts as an

on-pathway oligomer related to the further polymerization into fibrils. A11-recognized Aβ oligomer has an anti-parallel β-sheet structure and acts as an off-pathway oligomer unrelated or less related to the further assembly. In MITOL-deleted APP/PS1 brain, off-pathway Aβ oligomers detected by A11 antibody were dominant and preferentially generated (Supplementary Fig. 4j). In contrast, on-pathway Aβ oligomers detected by OC antibody were accumulated at the same levels regardless of MITOL expression in the brain (Supplementary Fig. 4k).

We finally investigated the causal relationship between the toxicity and the higher seeding effect of Aβ fibrils derived from the MITOL-deleted APP/PS1 brain. Upon the exposure of cells to Aβ fibrils isolated from the APP/PS1 brain, there were tendencies for increases in cell death and caspase-3/7 activity only in combination with free Aβ monomers (Fig. 4e, f), suggesting that the Aβ fibrils themselves do not exert neurotoxicity and that they enhance the toxicity of free Aβ monomers. Prominent induction of cell death and caspase-3/7 hyper-activation were observed upon treatment with free Aβ monomers and Aβ fibrils isolated from the MITOL-deleted APP/PS1 brain compared with those isolated from the APP/PS1 brain (Fig. 4e, f). These findings suggest that MITOL restricts the seeding effect of Aβ fibrils in order to attenuate the secondary assembly of free Aβ monomers. We next examined whether the worsening brain impairments in MITOL-deleted APP/PS1 mice are triggered by the toxicity of soluble Aβ oligomers. Previous studies defined rifampicin (RFP), a well-known antibiotic, as preventive medicine for Aβ pathology by using not only cellular models but also mouse models of Aβ pathology[40,41]. RFP inhibits Aβ oligomerization and oligomeric Aβ-dependent pathogenesis in both cellular and mouse models of Aβ etiology. Consistent with the previous studies, in this study, RFP injection reduced the accumulation of soluble Aβ oligomers in MITOL-deleted APP/PS1 mice (Supplementary Fig. 5a, b). Therefore, RFP is assumed to attenuate Aβ pathogenesis induced by the oligomeric forms of Aβ in MITOL-deleted APP/PS1 mice. As expected, there was a recovery of the severe loss of synaptophysin and microglial hyperactivation in MITOL-depleted APP/PS1 mice upon RFP administration (Fig. 4g, h, Supplementary Fig. 5c). The Barnes maze test also indicated that RFP rescued the exacerbated cognitive impairment in MITOL-deleted APP/PS1 mice (Fig. 4i). The locomotor activity of mice was not altered by RFP treatment (Supplementary Fig. 5d). Taken together, these results suggest that MITOL deletion accelerates Aβ pathogenesis in an Aβ oligomer-dependent manner.

In conclusion, the morphological perturbation of mitochondria, triggered by MITOL deletion, elevated the seeding activity of Aβ fibrils, resulting in the expansion of a fibrillized core in the plaques and the excessive secondary generation of dispersible off-pathway Aβ oligomers. The hyper-activation of Aβ secondary assembly catalyzed by Aβ fibrils led to more severe neurological

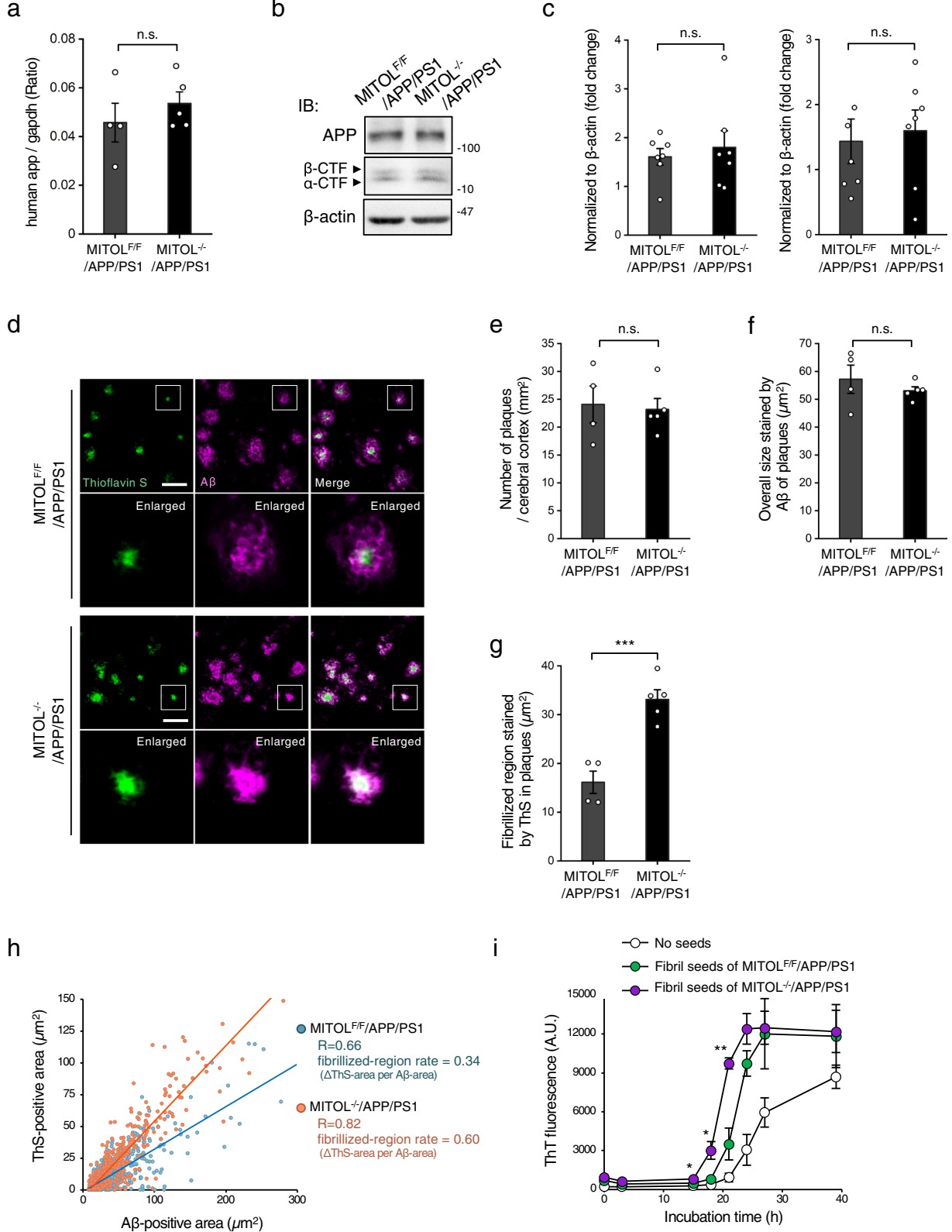

and behavioral symptoms in an oligomeric Aβ-dependent manner (Supplementary Fig. 5e).

## Discussion

**The seeding effect of Aβ fibrils and plaques in vivo**. AD is the most common neurodegenerative disorder and is caused by Aβ aggregates, including both the fibril form and the oligomeric

form. At present, Aβ amyloidosis is considered to follow the model whereby free Aβ monomers are spontaneously assembled and polymerized into fibrils through the process of oligomerization. On the other hand, recent studies suggest that the formed Aβ fibrils exert seeding activity by reducing the kinetic barrier for the aggregation of free Aβ monomers[32,33]. Therefore, the formation of Aβ fibrils and oligomers is determined by the balance

**Fig. 3 MITOL deletion elevates the seeding activity of Aβ plaques. a–c** MITOL deletion did not alter APP processing. mRNA was extracted from the hippocampus of indicated mice at 15 months of age and then analyzed by qRT-PCR (**a**). The hippocampal lysates of indicated 15-months old mice were immunoblotted with indicated antibodies (**b, c**). Error bars indicate ±SE (**a**: $n = 4$–5, **c**: $n = 7$). n.s.: not significant (Student's $t$ test). **d–h** MITOL deletion induced the formation of high-dense Aβ plaques with extended fibrillized core. Brain sections from indicated mice at 15 months of age were immunostained with anti-Aβ antibody 6E10 and Thioflavin S (ThS) (**d**). The lower panels show high-magnification images of the boxed regions. Scale bar represents 50 μm. The number and area of the overall region (Aβ-positive area), in addition to the size of the ThS-positive area, were calculated using ImageJ (**e–g**). Error bars indicate ±SE (**e–g**: $n = 4$–5). ***$p < 0.001$, n.s.: not significant (Student's $t$ test). **h** The fibrillized region in Aβ plaque was extended by MITOL deletion. The anti-Aβ antibody 6E10-stained area and ThS-stained area of each Aβ plaque were plotted in the scatter plot. Spearman's rank correlation coefficient ($R$) indicates the strength of a relationship between the total area and the cored area of Aβ plaque ($n =$ over 300). The rate of core region in Aβ plaques was represented by the slope of the best-fitting line for the scatter plot. **i** Aβ fibrils containing the plaques from the MITOL-deleted APP/PS1 brain exhibited a dramatic seeding effect on the polymerization of free Aβ monomers. Aβ fibrils in indicated 15-months old mice were isolated and purified as described in "Method" . The fibrils were co-incubated with seed-free Aβ40 monomers as aggregation seeds for indicated periods, as described in "Method", and the polymerization was monitored by ThT. As a control, TBS solution was used (no seeds). Error bars indicate ±SE ($n = 4$). **$p < 0.01$, *$p < 0.05$ (Student's $t$ test).

between self-assembly and the fibril-catalyzed secondary assembly. We report here that MITOL deletion increased the seeding activity of Aβ fibrils, thus inducing the formation of a highly dense fibrillized core in the plaques (Fig. 3d–i, Supplementary Fig. 3c–g). The morphological features of the Aβ plaques with high density are similar to those of another Aβ plaque-type, termed compact plaque, formed by plaque compaction and loss of the peripheral rim[42]. However, the Aβ plaques with a highly dense and larger fibrillized core in the MITOL-deleted APP/PS1 brain are clearly formed by extending the fibrillized region, unlike compact plaque, since the overall plaque size did not change regardless of MITOL expression (Fig. 3f). Numerous studies have demonstrated that secondary assembly is a more efficient and dominant reaction for newly generating Aβ aggregates from free Aβ monomers compared with self-assembly. Therefore, altering the seeding effect of Aβ fibrils (and also fibrillized plaques) is considered to be a critical factor for AD-related Aβ pathology, rather than the undirected and spontaneous formation of Aβ fibrils by self-assembly. Interestingly, MITOL deletion resulted in elevated Aβ43 in the Aβ fibrils (Supplementary Fig. 3h). At present, it is unclear whether the increase in Aβ43 is responsible for the enhanced seeding activity of Aβ fibrils. A previous report indicated that the Aβ species ratio, at least regarding the Aβ42/Aβ40 ratio, during self-assembly affects the fibril morphology[43]. Aβ fibrils were reported to catalyze the secondary assembly of free Aβ monomers on their surface, suggesting that the morphological differences of Aβ fibrils directly reflect the catalytic activity for secondary assembly[33]. Taking these findings together, enhanced secondary assembly of Aβ fibrils in the MITOL-deleted APP/PS1 brain might result from an imbalance of Aβ43 ratio in Aβ fibrils, which induces a morphological change of Aβ fibrils. Further studies to investigate the regulatory mechanisms of seeding by Aβ fibrils and plaques are needed.

**The dominant formation of toxic Aβ oligomers mediated by secondary assembly.** Several reports have indicated that both Aβ fibrils and oligomers are deleterious forms of Aβ. Since exogenous Aβ oligomers synthesized in vitro or isolated from brains with AD have been utilized to demonstrate severe toxicity, little is known about the regulatory mechanisms underlying the generation of toxic Aβ oligomers derived from endogenous and intact Aβ monomers. Enhanced self-assembly is clearly one of the mechanisms generating Aβ oligomers; however, this cannot explain why AD patients sometimes exhibit the dominant accumulation of Aβ oligomers with less conversion into Aβ fibrils and plaques. In the current model for Aβ amyloidosis, Aβ oligomers are considered to be spontaneously polymerized and subsequently converted into Aβ fibrils and plaques by their self-assembly activity; hence, increasing Aβ oligomers leads to

increasing Aβ fibrils and plaques. Consistent with this, diverse studies have indicated that enhancing the self-assembly of Aβ in vivo causes the accumulation of Aβ oligomers, fibrils, and plaques[44–46]. We showed here that MITOL deletion led to the preferential accumulation and toxicity of Aβ oligomers, without increasing the spontaneous formation of Aβ plaques (Figs. 3 and 4, Supplementary Fig. 3–5). Thus, the phenotype of Aβ amyloidosis in MITOL-deleted APP/PS1 mice, exhibiting an increase of Aβ oligomers but not of Aβ plaques, is unique and distinct from that of other gene transgenics/knockout mice with the enhancement of the self-assembly of Aβ. We also showed that the higher seeding activity of Aβ fibrils in the MITOL-deleted APP/PS1 brain strongly provoked the generation of toxic Aβ oligomers via the secondary assembly of free and dispersible Aβ monomers (Fig. 4a–d and Supplementary Fig. 4a, b). Indeed, the Aβ fibrils derived from the MITOL-deleted APP/PS1 brain robustly facilitated the formation of Aβ oligomers from free Aβ monomers compared with those derived from the APP/PS1 brain (Supplementary Fig. 4g), while no obvious difference in the acceleration of Aβ fibril formation was observed between them (Supplementary Fig. 3g). These findings imply that the secondary assembly by Aβ fibrils is a key mechanism underlying the dominant generation and accumulation of toxic dispersible Aβ oligomers, rather than the self-assembly of Aβ. This could explain why there are sporadic/familial AD patients mainly accumulating toxic Aβ oligomers without the enhanced formation of Aβ fibrils and plaques. According to our hypothesis, the Aβ pathology in AD can be classified into at least two types based on the primary alteration of Aβ amyloidosis: (1) patients with AD carrying abnormalities in Aβ self-assembly show the accumulation and toxicity of both Aβ oligomers and fibrils, and (2) other patients with AD with primary abnormalities in Aβ secondary assembly preferentially generate toxic Aβ oligomers, without (or with less) sequential polymerization into fibrils and plaques, and exhibit Aβ pathology in an oligomeric form-dependent manner.

We cannot fully explain why the enhancement of Aβ secondary assembly leads to the unilateral accumulation of Aβ oligomers with less conversion to fibrils. Some conformational changes causing Aβ oligomerization are reported to enhance the resistance to the subsequent aggregation into fibrils[47]. Based on this previous finding, the oligomerization of Aβ catalyzed by Aβ fibrils might accompany the generation of the off-pathway Aβ oligomers unrelated to the polymerization and fibrillization. MITOL deletion in the APP/PS1 brain indeed increased a major type of off-pathway oligomers detected by the A11 antibody (Supplementary Fig. 4h). Aβ fibrils containing a higher level of Aβ43, like those derived from the MITOL-deleted APP/PS1 brain (Supplementary Fig. 3h), might induce secondary assembly with

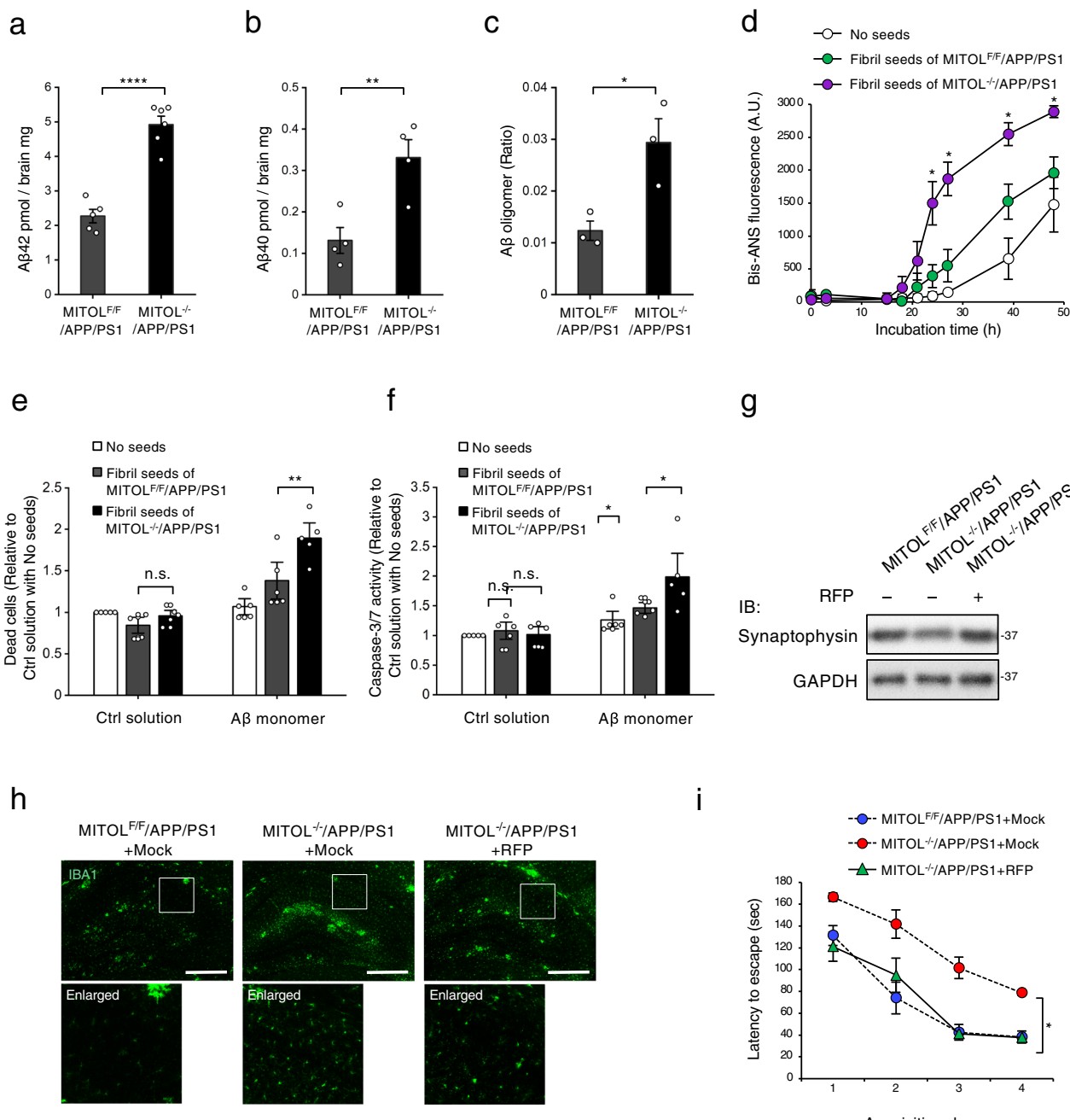

**Fig. 4 MITOL deletion increases toxic Aβ oligomer partly via the enhanced seeding activity of Aβ fibrils, leading to a higher neuronal injury.**
**a–c** MITOL deletion increased soluble Aβ oligomers. The TBS-soluble fraction was isolated from the cerebral cortex of indicated mice at 15 months of age to performed ELISA measurement of Aβ42 (**a**), Aβ40 (**b**), and Aβ oligomers (**c**). Error bars indicate ±SE (**a–c**: $n = 3$–6). ****$p < 0.0001$, **$p < 0.01$, *$p < 0.05$ (Student's $t$ test). **d** Aβ fibrils containing the plaques from the MITOL-deleted APP/PS1 brain exhibited a dramatic seeding effect on the oligomerization of free Aβ monomers. Aβ fibrils in indicated 15-months old mice were isolated and purified as described in "Method". The fibrils were co-incubated with seed-free Aβ40 monomers as aggregation seeds for indicated periods as described in "Method" and the oligomerization was monitored by Bis-ANS. As a control, TBS solution was used (no seeds). Error bars indicate ±SE ($n = 5$). **$p < 0.01$, *$p < 0.05$, n.s.: not significant (Student's $t$ test). **e**, **f** Aβ fibrils from MITOL-delated APP/PS1 boosted the neurotoxicity of free Aβ monomers. Purified amyloids of indicated mice were co-incubated with or without seed-free Aβ40 monomers for 24 h as described in "Method". These co-incubated solutions were added into the growth medium of SH-SY5Y cells at the 25 μM final concentration of Aβ40. After 48 h, dead cells were detected by LDH release assay (**e**) and Caspase-Glo 3/7 assay, respectively (**f**). Error bars indicate ±SE ($n = 5$). *$p < 0.05$, n.s.: not significant (Student's $t$ test). **g–i** RFP injection ameliorated the severe neurological and cognitive symptoms in MITOL-deleted APP/PS1 mice. RFP was treated as described in "Method". As a control, the solvent for RFP was also treated (Mock) (**g–i**). The hippocampus of indicated mice at 15 months of age was analyzed by immunoblotting (**g**) and immunostaining (**h**), respectively, using indicated antibodies. Barnes maze test was performed to evaluate learning memory of indicated mice at 15 months of age (**i**). Error bars indicate ±SE ($n = 6$–7). *$p < 0.05$ (Student's $t$ test).

the directed generation of toxic and off-pathway oligomers. Further studies are needed to define the interesting possibility that Aβ species ratios in the fibrils reflect the structural difference of Aβ oligomers generated by the fibril-mediated secondary assembly.

**The pathological relevance of mitochondrial pathophysiology in Aβ amyloidosis.** Previous reports have presented conflicting findings on the effects of mitochondrial pathophysiology on Aβ amyloidosis in vivo[26–28]. In this study, we investigated the detailed alterations of Aβ amyloidosis using model mice induced to exhibit morphological abnormalities of mitochondria by MITOL deletion. Accumulating reports have shown that deficiency of mitochondrial factors directly mediating mitochondrial fission/fusion, such as dynamin-related protein 1 (Drp1) or mitofusins, impairs mitochondrial dynamics without any distinction between physiological and pathological responses[12]. Therefore, the gene deletion of Drp1 or mitofusins in neurons leads to severe structural and functional impairments in the brain at the developmental stage[48,49]; hence, it is difficult to evaluate the causal relationship between mitochondrial morphology and Aβ amyloidosis in vivo. On the other hand, MITOL acts as a comprehensive regulator of these mitochondrial fission/fusion-related genes[14–18] and thus is available to examine the causal relationship between organized mitochondrial networks and aging-related pathology such as Aβ amyloidosis. Here, we suggest the novel insight that mitochondrial pathophysiology contributes to the expansion of the fibrillized core in Aβ plaques, rather than spontaneous Aβ plaque formation itself (Fig. 3 and Supplementary Fig. 3). This might explain why mitochondrial impairments lead to conflicting observations about Aβ plaques, based on the particular method used to detect such plaques. In fact, the fibrillized Aβ plaques in the MITOL-deleted APP/PS1 brain increased only the staining area of ThS, mainly reflected by fibrillized regions in Aβ plaques, without changing the staining area of anti-Aβ antibody detecting the overall regions of Aβ plaques (Fig. 3d–g). In addition, MITOL deletion significantly decreased the amount of non-toxic Aβ plaques without any fibrillized regions stained by only anti-Aβ antibodies (Supplementary Fig. 3b).

Numerous studies have suggested that multiple abnormalities of mitochondria affect Aβ amyloidosis itself[22–25], although the causal relationship between intracellular mitochondria and extracellular Aβ plaque formation remains poorly understood. We also cannot yet fully explain the molecular mechanisms connecting the higher seeding activity of Aβ fibrils and the mitochondrial pathophysiology triggered by MITOL deletion. Importantly, several reports have proposed an interesting idea that neurons provide the primary seed for initiating the formation of extracellular Aβ plaques by releasing intraneuronal Aβ microfibrils. Mitochondrial malfunction might alter the morphology or co-factors of intraneuronal Aβ microfibrils, rather than extracellular Aβ fibrils in the Aβ plaques. MITOL deletion at least altered the Aβ43 ratio in the fibrils (Supplementary Fig. 3h). It was previously reported that Aβ43 exerts a higher propensity to aggregate than other Aβ species[50], although the metabolic regulation of Aβ43, including its production and clearance, remains largely unknown. Several reports have proposed that γ-secretase and APP partly localize at mitochondria and mitochondrial contacts with other organelles, especially the ER[51,52]. Therefore, the specific γ-secretase and APP under the control of mitochondria might play pivotal roles in the metabolism or amyloidosis of Aβ43. Understanding the link between the neuronal primary seed and Aβ43 enrichment/mitochondrial function should lead to the elucidation of Aβ pathology and provide a new approach for treating AD with Aβ etiology. We indicated here that the downregulation of MITOL exacerbated Aβ pathology in a toxic Aβ oligomer-dependent manner and, in addition, that severe dementia symptoms triggered by MITOL repression were drastically rescued by RFP treatment. Therefore, the expression level of MITOL or the progression of mitochondrial pathophysiology may be available to classify the Aβ pathology, enabling the matching of effective drugs to patients with AD.

## Methods

**Reagents and antibodies.** Anti-MITOL rabbit polyclonal antibody was described previously[14]. Anti-APP C-terminal, anti-α-tubulin, and anti-β-actin antibodies were from Sigma. Anti-Aβ (D54D2), anti-synaptophysin, and anti-presenilin 1 (PS1) antibodies were from Cell Signaling. Anti-normal mouse IgG and anti-Tom20 antibodies were from Santa Cruz Biotechnology. The anti-PSD-95 antibody was from Abcam. Anti-Aβ (6E10) was from Covance. Anti-human Aβ E22P (11A1) antibody was from IBL. A11, OC antibodies were from Thermo Fisher Scientific. Anti-Iba1 was from Wako. Bis-ANS was from Cayman Chemical Company. DAPT and Aβ40 were from PEPTIDE INSTITUTE, INC. Aβ40 was dissolved in 0.02% ammonia solution at 300 μM. To obtain seed-free Aβ40 monomers, these Aβ solutions were centrifuged at $200,000 \times g$ for 3 h at 4 °C and the supernatants were collected. Thus, 0.02% ammonia solution was used as a control in the experiments using either Aβ40. ThS was from Sigma. ThT and Rifampicin were from Wako. Rifampicin was used according to a previous report[40]. Thus, 0.5% low-viscosity carboxyl methylcellulose (purchased from Sigma) was treated as control.

**DNA constructs and generation of cells stably expressing APPswe.** APPswe expression vector was constructed by mutagenesis of APPswe/ind expression vector (purchased from Addgene) and subcloning into pCX4 retrovirus vector[53]. SH-SY5Y cell lines stably expressing APPswe were generated using a retroviral expression system. Briefly, Plat-E cells were transfected with the pCX4 vector-encoding APPswe and pCMV vector-encoding VSV-G (purchased from Addgene) using lipofectamine LTX reagent with PLUS reagent (Invitrogen). After 48 h, the growth medium containing the retrovirus was collected. SH-SY5Y cells were incubated with the collected retrovirus-containing medium with 8 μg/mL polybrene (Sigma) for 24 h. After the selection by puromycin, SH-SY5Y stably expressing APPswe cell clones were selected by serial dilution.

**Cell culture and transfection.** SH-SY5Y cells were grown in Dulbecco's modified Eagle's medium (DMEM) supplemented with 10% fetal bovine serum (FBS) and penicillin/streptomycin. Cells were transfected with Lipofectamine 3000 (Invitrogen) according to the manufacturer's protocol. For siRNA transfection, Lipofectamine RNAiMAX (Invitrogen) was used according to the manufacturer's protocol. siPS1 and siMITOL were purchased by QIAGEN.

**Oxygen consumption rates (OCRs) measurement.** Mitochondrial OCR was measured with a Seahorse XFp Analyzer according to the manufacturer's protocol. Cells were in XF base Medium (containing 1 mM sodium pyruvate, 10 mM glucose, and 2 mM glutamine) at a density of 30,000 cells/well with combined reverse-transfection. The growth medium was replaced with an XF base medium (containing 1 mM sodium pyruvate, 10 mM glucose, and 2 mM glutamine) 1 h before assay. OCR was measured after sequentially exposure with 1 mM oligomycin, 0.5 mM FCCP, 0.5 mM rotenone, and antimycin A (Seahorse XF Cell Mito Stress Test Kit). OCR values were calculated with normalization to protein amount. Non-mitochondrial OCR has calculated the average of the OCR values after treatment with rotenone and antimycin A. Basal OCR was calculated from the OCR values before any chemical treatment subtracted by non-mitochondrial OCR. Maximal OCR was calculated from the ORC values after FCCP treatment subtracted by non-mitochondrial OCR.

**Mice.** MITOL$^{F/F}$ mice[19] were crossed with Emx1-Cre mice[54]. APP/PS1 (APPswe/PS1dE9) double-transgenic mice were obtained from The Jackson Laboratory (MMRRC Stock No. 034832). Genotype was confirmed by tail tipping mice at around 1 month of age. Mice were genotyped using the following PCR primers: Cre; forward: 5′–GTT TCA CTG GTT ATG CGG CGG–3′, reverse: 5′–TTC CAG GGC GCG AGT TGA TAG–3′, IL-2 (loading control); forward: 5′– CTA GGC CAC AGA ATT GAA AGA TCT–3′, reverse: 5′–GTA GGT GGA AAT TCT AGC ATC ATC C–3′, MITOL flox; forward: 5′–CAC AGG TAC GGT AGG TGT GTA AGC–3′, reverse: 5′–ATG GGA ATG TGG TTC AGT TGT ACC–3′, human APP; forward: 5′–GAA TTC CGA CAT GAC TCA GG–3′, reverse: 5′–GTT CTG CTG CAT CTT GGA CA–3′, human PSEN1; forward: 5′–AAT AGA GAA CGG CAG GAG CA–3′, reverse: 5′–GCC ATG AGG GCA CTA ATC AT–3′. All animals were maintained under university guidelines for the care and use of animals. The experiments were performed after securing Tokyo University of pharmacy and life sciences animal use committee protocol approval.

**Barnes maze test**. Barnes maze test was carried out according to established methods using a circular platform (90 cm diameter) with equally spaced 12 holes (5 cm diameter). Four visual cues is prepared in the surround of the apparatus. One of the 12 holes on the apparatus is attached to an escape box. The position of the escape box is fixed relative to visual cues, although the escape box connects a different hole every day. To conduct a trial for the banes maze test, a mouse was placed and habituated in the escape box. After 1 min, the mouse put in a blind black chamber located in the center of the apparatus. After 1 min, the chamber was lifted and the mouse was allowed to explore the circular platform with holes. A trial was recorded either up to the time when mouse entries into the escape box or 3 min elapses. Such trial was repeated three times per day as training. The latency to the escape box was averaged over three trials in a day.

**Novel object recognition test**. Novel object recognition test was carried out as described previously[55]. Briefly, the mouse was placed in the center of the open-filed box (40 cm long, 25 cm high, and 40 cm wide). habituation was performed for 10 min. After 24 h, the mouse was put in the center of the same box with two identical objects #A, 5 cm away from walls. Mouse movements were monitored either up to the time to reach the 20 s of total exploration or 10 min elapses. After 24 h, the mouse was placed in the center of the same box with two types of objects, a familial object #A and a novel object #B. Total exploration time was recorded as similar to training. The percentage of exploring time against a novel object #B was calculated.

**Y-Maze test**. The apparatus consists of three equal closed arms (each arm was 40 cm long, 14 cm high and 7 cm wide) converging to an equal angle. Each mouse is placed at the end of one arm and allowed to freely move through the maze for 10 min. An alternation is defined as consecutive entries in all three arms. The percentage of alternation was calculated.

**Open-field test**. Locomotor activity was measured using an open-field box. Mice were placed in the center of the apparatus, and mouse movements were recorded with a video camera for 5 min. The total distance traveled was calculated using Meander.

**Brain protein extraction**. To obtain a TBS-soluble fraction or amyloid-enriched fraction. The brain was extracted as described previously[34,56]. Briefly, the cerebral cortex of mice was homogenized in TBS buffer (10 mM Tris-HCl pH 7.4, 0.14 M NaCl, and protease inhibitors). Homogenates were centrifuged at $100,000 \times g$ for 1 h at 4 °C and the supernatants were collected as TBS-soluble fraction or non-amyloid fraction. The pellet was solubilized using 2% SDS buffer (25 mM Tris-HCl pH7.4, 2% SDS, and protease inhibitors) and centrifuged at $175,000 \times g$ for 2 h. The obtained pellet was extracted using TBS buffer as an amyloid (-enriched) fraction. The total amount of the isolated amyloid fraction was estimated by Aβ42-specific ELISA after the fraction was solubilized by 70% formic acid (see below).

**ELISA quantification**. Aβ40 and Aβ42-specific ELISA were performed with human β amyloid ELISA Kit, high sensitive (Wako) according to the manufacturer's protocol. Aβ oligomer-specific ELISA using 24B3 antibody was performed with a human amyloidβ toxic oligomer assay kit (IBL) according to the manufacturer's protocol. Other types of Aβ species was detected by ELISA using human amyloid β (1-x) assay kit (IBL), human amyloid β (1–38) (FL) assay kit (IBL), and human amyloid β (1–43) (FL) assay kit (IBL) according to the manufacturer's protocols, respectively. Conformation-specific ELISA, using A11 or OC antibody, for Aβ oligomer was performed according to sandwich ELISA procedure using an anti-Aβ antibody (6E10).

**Histology and immunohistochemistry**. For immunofluorescence analysis, brains were fixed in 4% PFA, and 20 μm slice sections were prepared. Slices were incubated for two nights at 4 °C with primary antibodies. Slices were incubated overnight with Alexa-conjugated secondary antibodies. The samples were analyzed using a FU1000-D confocal fluorescence microscope (Olympus) for Aβ plaque morphology or BZ-9000 fluorescence microscope BZ-9000 (KEYENCE) for others. ThS staining was performed at a concentration of 0.15% with twice wash using 80% ethanol. Analysis of the number, size of the intensity of Aβ plaques and Iba1-positive area was performed using ImageJ with plugins. Each image stained by indicated antibodies normalized and automatic thresholding based on the entropy of the histogram, followed by quantitation using the analyzed particles of ImageJ. For Aβ plaques, three different regions (3.5 mm²) in the cerebral cortex were analyzed in each slice. For microglia, the whole hippocampal region in each slice was analyzed. The quantification of the morphology of the Aβ plaques was performed as described previously[30,31]. Briefly, each image stained by ThS was normalized and thresholded at a value of 30 based on the entropy of the histogram. The modified images merged with anti-Aβ antibody 6E10 were used for the classification of Aβ plaques into two types based on the characterization of each Aβ plaque; (1) toxic Aβ plaque which exhibits not only Aβ-signal but also ThS-signal, (2) non-toxic Aβ plaque which displays only Aβ-signal without ThS signal more than 50 values. For nissl staining, brains were fixed with 10% formalin neutral buffer solution (Wako) and 5 μm-thick paraffin sections were prepared, followed

by staining with 0.5% cresyl violet. For cytochrome c oxidase (COX) histochemical staining, fresh-frozen sections were obtained and stained by cytochrome c oxidase staining solution (0.5 mg/ml diaminobenzidine, 1.3 mg/ml cytochrome c, and catalase) for 15 min in a humid chamber. After staining, the sections were rinsed four times for 10 min and dehydrated with increasing ethanol concentrations (70% for 2 min, 95% for 2 min, 99.5% for 10 min). The intensities of COX staining was quantified by subtracting off of the intensity in negative controls, as background signals, using ImageJ.

**Transmission electron microscopy (TEM)**. Mice brains were fixed by cardiac perfusion using 4% PFA and 2.5% glutaraldehyde, and 1 mm slice sections were prepared using Microslicer™ DTK-3000W in order to generate the 1 mm³ blocks. Blocks were incubated at 4% PFA and 2.5% glutaraldehyde for 24 h at 4 °C. These blocks were embedded in Epoxy resin and prepared as thin sections, followed by staining with 1% uranyl acetate and lead citrate. These sections were examined with an HT-7700 TEM (HITACHI, Tokyo, Japan).

**Immunoblotting**. Brain lysates were separated by SDS-PAGE and transferred to the PVDF membrane (Millipore). The blots were probed with the indicated antibodies, and protein bands on the blot were visualized by the enhanced chemiluminescence reagent (Millipore).

**RNA isolation and qRT-PCR**. Total RNA was isolated from the hippocampus of mice or cells using RNeasy kit (Qiagen) and subjected to reverse transcription to cDNA using ReverTra Ace qPCR RT Kit (TOYOBO) following the manufacturer's protocol. PCR was performed using a THUNDERBIRD SYBR qPCR Mix (TOYOBO). The PCR conditions were as follows: 95 °C for 1 min followed by 40 cycles at 95 °C for 10 s, 60 °C for 20 s and 72 °C for 30 s. RT-PCR was performed using mScript SYBR Green PCR kit and mScript primer assays (Qiagen). The following primers were used: mouse MITOL; forward: 5′–TTC ACC AGG CTT GTC TCC A–3′, reverse: 5′–GCA TCA CTG TCA CTG CTC CA–3′, human MITOL; forward: 5′–GAG CAG TGA CAG TGA TGC AG–3′, reveres: 5′–AGG ACA ACC TAT CCC TGG AA–3′, GAPDH; forward: 5′–CCT GCA CCA CCA ACT GCT TAG C–3′, reverse: 5′–GCC AGT GAG CTT CCC GTT CAG C–3′, CD68 forward: 5′–AGC CTT GTG TTC AGC TCC AAG CCC–3′, reverse: 5′–ATG CCC CAA GCC CTC TTT AAG CCC–3′, IL-1β: forward: 5′–GGA CCC ATA TGA GCT GAA AGC–3′, reverse: 5′–TCG TGG CTT GGT TCT CCT TGT–3′, IL-6; forward: 5′–TGG AGT CAC AGA AGG AGT GGC TAA G–3′, reverse: 5′–CTG ACC ACA GTG AGG AAT GTC AA–3′, TNF-α; forward: 5′–CAT CAG TTC TAT GGC CCA GA–3′, reverse: 5′–TGC TCC TCC ACT TGG TGG TT–3′, human APP; forward: 5′–GAA TTC CGA CAT GAC TCA GG–3′, reverse: 5′–GTT CTG CTG CAT CTT GGA CA–3′, IDE; forward: 5′–ACT AAC CTG GTG GTG AAG −3′, reverse: 5′–GGT CTG GTA TGG GAA ATG–3′, NEP; forward: 5′–GCA GCC TCA GCC GAA ACT AC–3′, reverse: 5′–CAC CGT CTC CAT GTT GCA GT–3′, MMP2; forward: 5′–GTC GCC CCT AAA ACA GAC AA–3′, reverse: 5′–GGT CTC GAT GGT GTT CTG GT–3′, MMP9; forward: 5′–CGT CGT GAT CCC CAC TTA CT–3′, reverse: 5′–AAC ACA CAG GGT TTG CCT TC– 3′

**In vitro mitochondrial ATP production assay**. Mitochondria were isolated as described previously[57]. These isolated mitochondrial fractions were subjected to the in vitro mitochondrial ATP production assay as described previously[58,59]. Briefly, isolated mitochondria were incubated in assay buffer (5 mM KCl, 1 mM K2HPO4, 1 mM MgCl2, 10 mM HEPES, 0.04 mM EGTA, 0.5 mg/mL BSA, 250 mM ADP, 1 mM pyruvate, 1 mM malate) for 15 min at 28 °C. After the reaction, the ATP amount was measured using the ATP luminescent assay kit (Abcam) following the manufacturer's instructions.

**Aβ aggregation assay using ThT, Bis-ANS, or oligomeric Aβ-specific antibody**. ThT fluorescence assay and Bis-ANS fluorescence assay with the addition of fibrils as aggregation seed was performed as described previously[32,35]. To analyze secondary-aggregation, stock seed-free Aβ40 monomer solution was diluted to 75 μM using TBS buffer including 5.5 μM ThT, 22.2 mM glycine-NaOH and incubated with or without 10 pM Aβ fibrils-enriched fraction at 37 °C. The measurement of secondary-oligomerization was performed using 5.5 μM Bis-ANS instead of ThT. When seed-free Aβ40 monomer solutions were incubated without any Aβ fibrils, TBS buffer was added as control (no seed). ThT or Bis-NAS fluorescence at emission at 482 or 523 nm, respectively, was monitored using granting microplate reader (CORONA). The lag-phase and elongation rate was calculated as described previously[60]. To check the oligomerization of Aβ40 monomers, stock seed-free Aβ40 monomer solution was diluted to 25 μM using TBS buffer including 22.2 mM glycine-NaOH and incubated with or without 3 pM Aβ fibrils-enriched fraction at 37 °C. After the indicated period, these solutions were collected and mixed with an equal amount of PBS, followed by dot blot with Aβ oligomer-specific antibody (11A1).

**Cell death assay**. To prepare Aβ aggregates including oligomers and fibrils, stock seed-free Aβ40 monomer solution was diluted to 150 μM using TBS buffer including 22.2 mM glycine-NaOH and incubated with or without 150 pM amyloid-enriched fraction at 37 °C for 24 h. In total, 0.02% ammonia solution was used as

control. These solutions including Aβ aggregates were added into growth DMEM with 1% FBS at the final concentration of 25 μM. SH-SY5Y cells were exposed for 48 h. To monitor the level of cell death induction, cytotoxicity LDH assay kit (Dojindo) or caspase-Glo 3/7 assay kit (Promega) was used according to each manufacturer's protocol. The activity of caspase-3/7 was normalized to the total protein amount of lysed samples.

**Statistics and reproducibility**. All results are expressed as mean ± SD in cellular experiments and mean ± SE in both mouse experiments and in vitro experiments, with the except for TEM analysis. In TEM analysis, the boxes represent the 25th and 75th percentiles, the line expresses as the median, and the whiskers represent the 10th and 90th percentiles. For behavioral tests, dot plots graphs were generated using the open-source web tool PlotsOfDate (https://huygens.science.uva.nl/PlotsOfData/)[61]. Obtained data were compared between independent experiments using either a two-tailed Student t-test or one-way ANOVA with Bonferroni's *post hoc* test followed by Tukey HSD test. The number of independent experiments is shown as *n*.

## Data availability

The unprocessed immunoblotting images are shown in Supplementary Fig. 6. The source data behind the graphs are provided in Supplementary Data 1. All datasets in the current study are available from the corresponding author on reasonable requests.

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

## Acknowledgements
We thank Erika Mochizuki, Toshifumi Fukuda, and Takeshi Tokuyama for their technical assistance. This study was supported in part by MEXT/JSPS KAKENHI (18H04869 and 20H04911 to S.Y.) and MEXT-Supported Program for the Strategic Research Foundation at Private Universities (to S.N., R.I., and S.Y.), The Uehara Memorial Foundation, The Naito Foundation (to S.N. and S.Y.), The Takeda Foundation, The Sumitomo Foundation, The Cosmetology Research Foundation, The Ono Medical Research Foundation, The Tokyo Biochemical Research Foundation, and AMED under Grant numbers 17gm5010002, 18gm5010002, 19gm5010002, and 20gm5010002 (to S.Y.).

## Author contributions
K.T. and S.Y. designed and carried out the experiments. A.U., H.I., S.N., I.S., N.I. and R.I. contributed to the understanding of MITOL-deleted APP/PS1 mice phenotype. K.T., S.N., and M.M. carried out TEM experiment and analysis. H.I. also carried out cultured cell analysis and RFP treatment. M.M. carried out Aβ aggregation assay using ThT, Bis-ANS, or 11A1 antibody and performed a cell death assay. M.M. provided us with APP/PS1 mice. K.T. and S.Y. wrote the manuscript with contributions from R.I.

## Competing interests
The authors declare no competing interests.
