## [Peer Review File · Communications Biology]

Reviewers' comments:

Reviewer #1 (Remarks to the Author):

Takeda and coworkers present a wonderful analysis of the role of MITOL in mitochondria(mt) dynamics and how it relates to Alzheimer disease (AD). The data is for the most part compelling that mt quality control through MITOL is critical for amyloid form, synapse related features, and learning. What is less clear is pushing the data to say the model is AD--no mouse model is AD, and that amyloid oligomers are at the formation of AD--still unclear. None of these points is essential for this to be an important study.

1.English could use minor improvement.

2.The whole issue of seeding and cores is complex. It would seem essential to sequence the amyloid in both cases +/- MITOL to see if there are differences in the protein. Additionally it could be the mechanism at a cellular level is different +/- MITOL. It is moving too far ahead to rely on seeding as the sole issue.

2.The synapse approach used is a marker and is not the same as synapses. This should be clear in the text. EM would be needed--I am fine with the marker data but it should not be stated as synapses as other papers have mislead for years.

4.The EM is excellent

5.Figure 1I is impossible to interpret -should be redone.

6.Get away from saying something protects/causes from AD based on this model.

Reviewer #2 (Remarks to the Author):

In this manuscript, the authors showed that MITOL depletion in AD model mice aggravates behavior deficits such as cognitive impairment. They suggested that MITOL depletion specifically enhanced the seeding effect of A β plaques, leading to toxic and soluble A β oligomers. I appreciate their enormous efforts on generation of MITOL-/- APP/PSI, and analysis of behavior changes and A β metabolism in these mice. However, I have a considerable concern about their logical flow and their overinterpretation. APP/PSI mice and cell models exhibited a significant reduction of MITOL expression. In this case, I think that authors should overexpress MITOL gene rather than depleting the MITOL in APP/PSI mice to see the effect of MITOL in AD pathogenesis. Because MITOL-/-APP/PSI mice showed the aggravated behavior and enhanced seeding effect of A β plaques, it is likely that MITOL affects AD pathogenesis in different pathways (not the same pathway) showing additive effect. Second, they don't provide any mechanism by which how MITOL E3 ligase affects A β metabolism. It would be very informative if they show some biochemical data in AD cells.

[Major points]

- In Fig. 1A-D, the authors showed a reduction of MITOL expression at the level of mRNA and protein in APP/PSI mice and cell models. In Fig. 1D. They showed that DAPT as a γ -secretase inhibitor, blocked a reduction of MITOL, interpreting that "A β decreases MITOL expression" in page 9. At least, they must show whether DAPT restored the MITOL protein level, too. I think that it is fair to just write a reduction of MITOL expression in APP/PSI models.

- Fig. 1E, MITOL expression in APP models (Fig. 1A-D) was already significantly reduced. Do authors claim that the remaining MITOL in APP mice shows protecting effect on mitochondria because MITOL-/- APP/PSI mice showed a severe phenotype on mitochondria phenotype.

- Fig.1I: Although authors showed staining patterns in the presence of respiratory inhibitors (Fig S1E) they must improve the resolution of SDH, COX staining with magnification. All the tissue sections, not the neuronal cells, showed the same staining patterns.

- In Fig 2, To simplify the data and their statistical analysis, it will be easier to first describe depletion of MITOL gene by itself does not cause any cognitive deficits (Figure S2A, S2B). Then compare the difference between MITOL F/F and MITOL F/F/APP/PSI to see that these mice exhibit AD phenotypes. Finally, AD phenotypes were worsened in the MITOL-/-/APP/PSI mice. In Fig.2C, correct the statistical labeling.
- In Fig.2F, I wonder whether overexpression of MITOL in AD cell model (Fig. 1C&D) change the expression levels of synaptophysin and PSD-95.
- In Fig. 2H, the authors just describe the neuroinflammatory phenotype of the MITOL-/- APP/PSI mice. The authors previously reported that neuron-specific MITOL-/- developed enhanced oxidative stress in brain, which led to microglial activation (Life Sci Alliance, 2019). How they reconcile with these findings? Whether APP/PSI mice do not develop any neuroinflammatory phenotypes? Is it consistent to the previous report?
- In Fig.3: Amyloid fibril indicator Ths stains the amyloid-core of each A β plaque. They found that the A β plaque in MITOL-depleted APP/PSI brain exhibited an extended amyloid-core (Fig. 3G&H). From these findings authors suggested that "MITOL facilitates the fibrilization in A β plaque (??) in page 14. Is it right? It is very confusing...
- In Fig.3&4. It will help if they add a working diagram of the sequential events of A β plaque formation and where MITOL probably functions.
- In Fig. 4G, using AD cell model, can authors show whether overexpression of MITOL in AD cell model (Fig. 1C&D) change the expression levels of synaptophysin?

[Minor points]

- Fig. 2A, please replace the labeled symbols with color like Fig.2B (minor)
- Fig. 3I, please replace the symbols with color.

Reviewer #3 (Remarks to the Author):

In this study, authors have investigated the role of MITOL in Alzheimer's disease progression. In this study authors have tried to connect two different phenomena in AD brain: mitochondrial dysfunction and A β -plaque formation. The study demonstrated that MITOL has a direct effect on mitochondrial dynamics and functioning. The proteins levels were shown to be reduced in AD model. Further authors have shown that MITOL ablation can enhance the seeding effect of A β -plaques and lead to cognitive decline in the transgenic mice model of AD.

This study mostly comprises in vivo results, thus carries an obvious weightage. The experimental designs in this study were found to satisfactory. The study can improve the understanding between mitochondria and A β oligomer mediated neuronal toxicity.

However, a few minor modifications are indicated below:

1. It should be noted that the study lacks an explanation how mitochondrial dysfunction and enhanced accumulation of A β oligomers are interconnected, or whether they are connected at all or not. Mitochondrial dysfunction seems to be via MITOL and the observations on A β oligomer accumulation could be a secondary effect due to mitochondrial dysfunction. The authors may at least discuss this point clearly and make an effort to point out which could be the causal factor and which one is the consequence.
2. Throughout the manuscript, language is not up to the mark. This needs to be thoroughly improved. A few sentences are mentioned below which should be considered for grammatical corrections or properly stated. The list is not exhaustive.
 - Therefore, the preferential accumulation of toxic A β oligomers with less capable of converting to

plaques might be a key etiology

- To avoid defects in mitochondrial functions, individual mitochondrion is habitually integrated by continuous remodelling through fusion and fission.
- In addition, it is not yet fully clarified how AD pathogenesis is altered by perturbation of mitochondrial dynamics and engaging mitochondrial pathophysiology.
- Most recently, we have generated neuron-specific MITOL knockout (KO) mice in the first time.....
- These were no obvious signals in COX or SDH staining...
- Likewise, MITOL deletion in APP/PS1 mice led to worsening impairments of the working memory....
- A β aggregates into oligomers and fibrils, generally accepted to trigger neuronal dysfunction and neuroinflammation in the brain of AD patients and AD mouse models before the....
- Disturbing mitochondrial bioenergetics in AD, designed through different experimental approaches, leads to inconsistency outcomes represented by increased or decreased A β
- In order to understand how mitochondrial malfunctions engages the alternation of A β pathology, there is a clear need to reappraise about A β state using AD model brain with mitochondrial impairments.....

- In contrast, on-pathway A β oligomers detected....

- We also evaluated whether the worsening AD pathology by MITOL deletion....

3. At first instance, please describe APP/PS1 mice with its full form and then use the short form for rest of the manuscript.

4. Consistent to the data obtained by APP/PS1 mice, MITOL..... please mention that this was done in cell line.

5. Figure 1E does not show any decrease in area, as claimed by the authors. Better representative image is required.

6. Authors may reconsider using the term "neuronal size" for figure 2D, as cresyl violet stains only the soma or cell body of neuron.

7. Authors are requested to re-evaluate the statistics for PSD95 quantification, in figure 2G,

8. A β staining is not visible in figure S4C, so it is not clear what the authors have quantified.

Referee #1

Takeda and coworkers present a wonderful analysis of the role of MITOL in mitochondria(mt) dynamics and how it relates to Alzheimer disease (AD). The data is for the most part compelling that mt quality control through MITOL is critical for amyloid form, synapse related features, and learning. What is less clear is pushing the data to say the model is AD--no mouse model is AD, and that amyloid oligomers are at the formation of AD--still unclear. None of these points is essential for this to be an important study.

We greatly appreciate your constructive comments, which have helped us to make our paper stronger. Particularly, I am not an expert in the field of Alzheimer's disease, so I inadvertently described our mouse as an AD model mouse. In the revised paper, I deleted this expression and replaced it with A β pathology. We believe that we could address almost all concerns.

1. English could use minor improvement.

The English in our manuscript has been checked by a company's service for English editing.

2. The whole issue of seeding and cores is complex. It would seem essential to sequence the amyloid in both cases +/- MITOL to see if there are differences in the protein. Additionally it could be the mechanism at a cellular level is different +/- MITOL. It is moving too far ahead to rely on seeding as the sole issue.

We appreciate this comment as it was a very important suggestion for our research. According to this comment, we performed A β species-specific ELISA assays in order to estimate the amount of each major A β species in the isolated amyloids/fibrils (Figure S3H). Interestingly, the amyloids/fibrils derived from the MITOL-deleted APP/PS1 brain contained more A β 43, compared to A β 40, than those derived from the APP/PS1 brain. A previous report has suggested that A β 42/A β 40 ratio in the self-assembly process affects the morphology of the formed fibrils. It was also reported that A β fibrils catalyze the secondary-assembly of free A β monomers on their surface, suggesting that the morphological differences of A β fibrils are directly reflected in the catalytic activity for secondary-assembly. Therefore, this result obtained following your nice advice raises an interesting possibility that an increased ratio of A β 43 to A β 42 in the fibrils might lead to the morphological change of A β fibrils, followed by enhancing the seeding effect. Although at present we cannot fully mentioned this interesting idea based on the increased A β 43 ratio to A β 40 or A β 42, we briefly mentioned it in a discussion that enhanced secondary assembly of A β fibrils in the MITOL-deleted APP/PS1 brain might result from an imbalance of A β 43 ratio in A β fibrils, which induces a morphological change of A β fibrils.

3. The synapse approach used is a marker and is not the same as synapses. This should be clear in the text. EM would be needed--I am fine with the marker data but it should not be stated as synapses as other papers have mislead for years.

We carefully corrected sentences explaining about synapse markers synaptophysin and PSD-95.

4. The EM is excellent

We are grateful for your understanding. We also consider that the EM is important data in order to mention the morphological alternation of mitochondria.

5. Figure 1I is impossible to interpret -should be redone.

We redid the COX staining with high-magnification images (Figure 1I, S1G). In addition, we performed an “in vitro mitochondrial ATP assay” in order to monitor the mitochondrial activity for ATP production (Figure 1J). Mitochondria in the MITOL-deleted APP/PS1 brain showed the impaired bioenergetics consistent with the result obtained by COX staining.

6. Get away from saying something protects/causes from AD based on this model.

We appreciate you to mention it. We are very sorry for describing our mouse as an AD model mouse inappropriately. In the revised paper, I deleted this expression and replaced it with A β pathology.

Referee #2

In this manuscript, the authors showed that MITOL depletion in AD model mice aggravates behavior deficits such as cognitive impairment. They suggested that MITOL depletion specifically enhanced the seeding effect of A β plaques, leading to toxic and soluble A β oligomers. I appreciate their enormous efforts on generation of MITOL^{-/-} APP/PS1, and analysis of behavior changes and A β metabolism in these mice. However, I have a considerable concern about their logical flow and their overinterpretation. APP/PS1 mice and cell models exhibited a significant reduction of MITOL expression. In this case, I think that authors should overexpress MITOL gene rather than depleting the MITOL in APP/PS1 mice to see the effect of MITOL in AD pathogenesis. Because MITOL^{-/-}/APP/PS1 mice showed the aggravated behavior and enhanced seeding effect of A β

We are very grateful to you for helpful comments. I absolutely agreed with your opinion that the transgenic mice overexpressing MITOL are very useful and important for this research.

To be honest, we had already tried to generate the mice overexpressing MITOL using two different experimental designs. However, we did not get any transgenic mice overexpressing MITOL in spite of introducing MITOL gene, because MITOL has a strong auto-ubiquitination activity and therefore it is rapidly degraded to the basal level by auto-clearance. Even in the cell expression system, MITOL can be only transiently overexpressed, but its expression level returns to the basal level in a few days. This is the reason that there are no stable cell lines overexpressing MITOL. We conclude that MITOL expression is tightly and strictly controlled in both in vitro and in vivo. I greatly appreciate your understanding about this matter.

Instead, we analyzed the alternation of A β amyloidosis using the MITOL-deleted APP/PS1 brain in detail. We clarified here that A β fibrils themselves, derived from

the MITOL-deleted APP/PS1 brain, potentially exert a higher activity for cell death induction regardless of MITOL expression in cells treated with A β fibrils (Figure 4E, 4F). The data suggests an important fact that the A β fibrils, formed in the MITOL-deleted APP/PS1 brain, have a stronger toxicity than those formed in the APP/PS1 brain. Thus, we consider that the primary-alteration in A β amyloidosis in the MITOL-deleted APP/PS1 brain is on the fibril formation, leading to the morphological changes of the fibrils followed by a higher seeding activity.

[Major points]

- In Fig. 1A-D, the authors showed a reduction of MITOL expression at the level of mRNA and protein in APP/PS1 mice and cell models. In Fig. 1D. They showed that DAPT as a γ -secretase inhibitor, blocked a reduction of MITOL, interpreting that "A β decreases MITOL expression" in page 9. At least, they must show whether DAPT restored the MITOL protein level, too. I think that it is fair to just write a reduction of MITOL expression in APP/PS1 models.

We agree with your concern. We added a new data showing that DAPT treatment restored the reduction of MITOL protein in APP^{swe}/siPS1 expressing cells (Figure S1B).

- Fig. 1E, MITOL expression in APP models (Fig. 1A-D) was already significantly reduced. Do authors claim that the remaining MITOL in APP mice shows protecting effect on mitochondria because MITOL^{-/-} APP/PS1 mice showed a severe phenotype on mitochondria phenotype.

According to your comment, we improved the sentence explaining Figure 1E as follows (p9-p10).

"We then analyzed individual mitochondrial morphology in vivo using transmission electron microscopy (TEM). In the APP/PS1 brain, MITOL deletion significantly increased the number of smaller mitochondria (Figures 1E, 1F, S1F). This indicates that the remaining expression of MITOL in the APP/PS1 brain (approximately 40% compared with the level in non-transgenic mice) still has a sufficient protective effect on mitochondrial morphology."

- Fig. 1I: Although authors showed staining patterns in the presence of respiratory inhibitors (Fig S1E) they must improve the resolution of SDH, COX staining with magnification. All the tissue sections, not the neuronal cells, showed the same staining patterns.

We tried to improve the staining method and obtained a clear image with high-magnification as shown in Figure 1I. In addition, we performed a new experiment, termed mitochondrial ATP production assay, in order to make up for changes of mitochondrial activity. Using the isolated mitochondria derived from each group of mice, we showed that MITOL deletion from the APP/PS1 brain resulted in the malfunction in mitochondrial ATP production. In consistent with the TEM image showing the disrupted mitochondrial cristae in the MITOL-deleted APP/PS1 brain, these data supported that MITOL plays a protective role against mitochondrial malfunctions in A β pathology.

- In Fig 2, To simplify the data and their statistical analysis, it will be easier to first describe depletion of MITOL gene by itself does not cause any cognitive deficits (Figure S2A, S2B). Then compare the difference between MITOL F/F and MITOL F/F/APP/PS1 to see that these mice exhibit AD phenotypes. Finally, AD phenotypes were worsened in the MITOL-/-/APP/PS1 mice. In Fig.2C, correct the statistical labeling.

We apologize that our explanation was not clear as the reviewer pointed out. We corrected the sentences about behavior tests (Figure 2A-2C) as follows (p11).

“The object recognition memory and spatial working memory were assessed by the novel object recognition test and Y-maze test, respectively, using 15-month-old mice (Figure 2A, 2B). MITOL deletion itself did not cause any defects in these memory functions, whereas APP/PS1 mice developed mild memory impairments (Figure 2A, 2B). MITOL-deleted APP/PS1 mice exhibited the most severe behavioral abnormalities among all groups of mice (Figure 2A, 2B). Likewise, the Barnes maze test revealed that MITOL deletion led to worsening cognitive decline in APP/PS1 mice (Figure 2C).”

- In Fig.2F, I wonder whether overexpression of MITOL in AD cell model (Fig. 1C&D) change the expression levels of synaptophysin and PSD-95.

We appreciate your suggestion. However, no synapse formation can be induced in SH-SY5Y cells even in any conditions including differentiation stimuli. Thus, we tested the effect of MITOL deletion on A β pathology mainly using *in vivo* model (i.g. Figure 2F, 2G).

- In Fig. 2H, the authors just describe the neuroinflammatory phenotype of the MITOL-/-/APP/PS1 mice. The authors previously reported that neuron-specific MITOL-/- developed enhanced oxidative stress in brain, which led to microglial activation (Life Sci Alliance, 2019). How they reconcile with these findings? Whether APP/PS1 mice do not develop any neuroinflammatory phenotypes? Is it consistent to the previous report?

We previously demonstrated that MITOL deletion in brain itself enhanced oxidative stress and induced microgliosis (Nagashima et al 2019, Life Sci Alliance). In this previous report, we mainly analyzed the total number of microglia using microglial marker Iba1. On the other hand, in the present study, we used other markers, such as CD68 and cytokines, in order to monitor the activation of microglia, because the inflammatory activation of microglia is more important in A β pathology, rather than the total number of microglia. As shown in Figure S2D and S2E, MITOL deletion itself led to an increase of the number of microglia, however, the microglial alteration only by MITOL deletion was not sufficient to provoke the subsequent production of cytokines. In the same age of mice, the APP/PS1 brain showed only a mild neuroinflammation including an increase of microglia and upregulation of TNF α . In contrast, MITOL deletion in the APP/PS1 brain significantly accelerated the inflammatory reaction, including upregulation of other markers IL-1 β , IL6.

- In Fig.3: Amyloid fibril indicator Ths stains the amyloid-core of each A β plaque. They found that the A β plaque in MITOL-depleted APP/PS1 brain exhibited an extended amyloid-core (Fig. 3G&H). From these findings authors suggested that “MITOL facilitates the fibrilization in A β plaque (??) in page 14. Is it right? It is very confusing...

We are sorry for our unclear explanations. We improved the sentences about amyloid and fibrillization.

-In Fig.3&4. It will help if they add a working diagram of the sequential events of A β plaque formation and where MITOL probably functions.

I appreciate your useful advice. We added a working diagram with the sequential events A β aggregation (Figure S3C) and improved our schematic model in order to help the understanding of A β plaque formation regulated by MITOL.

-In Fig. 4G, using AD cell model, can authors show whether overexpression of MITOL in AD cell model (Fig. 1C&D) change the expression levels of synaptophysin?

We appreciate your suggestion. However, no synapse formation can be induced in SH-SY5Y cells even in any conditions including differentiation stimuli. Thus, we tested the effect of MITOL deletion on A β pathology mainly using *in vivo* model (i.g. Figure 2F, 2G).

[Minor points]

- Fig. 2A, please replace the labeled symbols with color like Fig.2B (minor)
- Fig. 3I, please replace the symbols with color.

We are sorry for our unclear labels. We changed the symbol colors to avoid misleads. Regarding Figure 3I (and 4D), we also added the bar graphs calculated from the line graphs in Figure 3I and 4D as the supplemental figures (Figure S3F, S3G, S4F, S4G).

Referee #3

In this study, authors have investigated the role of MITOL in Alzheimer's disease progression. In this study authors have tried to connect two different phenomena in AD brain: mitochondrial dysfunction and A β -plaque formation. The study demonstrated that MITOL has a direct effect on mitochondrial dynamics and functioning. The proteins levels were shown to be reduced in AD model. Further authors have shown that MITOL ablation can enhance the seeding effect of A β -plaques and lead to cognitive decline in the transgenic mice model of AD. This study mostly comprises *in vivo* results, thus carries an obvious weightage. The experimental designs in this study were found to satisfactory. The study can improve the understanding between mitochondria and A β oligomer mediated neuronal toxicity. However, a few minor modifications are indicated below:

First of all, we apologize for many grammatical mistakes in our manuscript.
We are very grateful to you for helpful suggestions and comments.

1. It should be noted that the study lacks an explanation how mitochondrial dysfunction and enhanced accumulation of A β oligomers are interconnected, or whether they are connected at all or not. Mitochondrial dysfunction seems to be via MITOL and the observations on A β oligomer accumulation could be a secondary effect due to mitochondrial dysfunction. The authors may at least discuss this point clearly and make an effort to point out which could be the causal factor and which one is the consequence.

At present, we cannot fully explain the molecular mechanisms connecting between the higher seeding activity of A β fibrils, triggering the excessive generation of A β oligomers, and mitochondrial pathophysiology by MITOL deletion. In our hypothesis, MITOL can alter the seeding activity of A β fibrils via changes of mitochondrial function. In the revised version, we obtained a new interesting data that A β fibrils in the MITOL-deleted APP/PS1 brain specifically increased A β 43 as their the composition, but not other species of A β (Figure S3H). Based on the finding, we discussed about molecular mechanisms connecting between A β fibril formation and mitochondrial pathophysiology as follows. (p29)

“Importantly, several reports have proposed an interesting idea that neurons provide the primary seed for initiating the formation of extracellular A β plaques by releasing intraneuronal A β microfibrils. Mitochondrial malfunction might alter the morphology or co-factors of intraneuronal A β microfibrils, rather than extracellular A β fibrils in the A β plaques. MITOL deletion at least altered the A β 43 ratio in the fibrils (Figure S3H). It was previously reported that A β 43 exerts a higher propensity to aggregate than other A β species⁵⁰, although the metabolic regulation of A β 43, including its production and clearance, remains largely unknown. Several reports have proposed that γ -secretase and APP partly localize at mitochondria and mitochondrial contacts with other organelles, especially the ER^{51,52}. Therefore, the specific γ -secretase and APP under the control of mitochondria might play pivotal roles in the metabolism or amyloidosis of A β 43.”

2. Throughout the manuscript, language is not up to the mark. This needs to be thoroughly improved. A few sentences are mentioned below which should be considered for grammatical corrections or properly stated. The list is not exhaustive.

- Therefore, the preferential accumulation of toxic A β oligomers with less capable of converting to plaques might be a key etiology
- To avoid defects in mitochondrial functions, individual mitochondrion is habitually integrated by continuous remodelling through fusion and fission.
- In addition, it is not yet fully clarified how AD pathogenesis is altered by perturbation of mitochondrial dynamics and engaging mitochondrial pathophysiology.
- Most recently, we have generated neuron-specific MITOL knockout (KO) mice in the first time.....
- These were no obvious signals in COX or SDH staining...
- Likewise, MITOL deletion in APP/PS1 mice led to worsening impairments of the working memory....
- A β aggregates into oligomers and fibrils, generally accepted to trigger neuronal dysfunction and neuroinflammation in the brain of AD
- A β aggregates into oligomers and fibrils, generally accepted to trigger neuronal dysfunction and neuroinflammation in the brain of AD patients and AD mouse models before the.....
- Disturbing mitochondrial bioenergetics in AD, designed through different experimental approaches, leads to inconsistency outcomes represented by increased or decreased A β
- In order to understand how mitochondrial malfunctions engages the alternation of A β pathology, there is a clear need to reappraise about A β state using AD model brain with mitochondrial impairments.....
- In contract, on-pathway A β oligomers detected....
- We also evaluated whether the worsening AD pathology by MITOL deletion....

We are very sorry for our poor English. The grammar in the revised paper was checked again and improved by a native speaker.

3. At first instance, please describe APP/PS1 mice with its full form and then use the short form for rest of the manuscript.

According to your suggestion, we corrected the sentences as follows.

“APP^{swe}/PSEN1^{ΔE9} transgenic mice, referred to here as APP/PS1 mice, are widely recognized as a mouse model for AD-related A β pathology. APP/PS1 mice contain the human transgene with the Swedish mutation (KM595/596NL) of APP (APP^{swe}) combined with a deletion mutation of exon 9 in PS1 (PS1 Δ E9).”

4. Consistent to the data obtained by APP/PS1 mice, MITOL..... please mention that this was done in cell line.

We improved our manuscript with obvious distinction between mice and cellular models.

5. Figure 1E does not show any decrease in area, as claimed by the authors. Better representative image is required.

Following your advice, we replaced them to better images (Figure 1F).

6. Authors may reconsider using the term “neuronal size” for figure 2D, as cresyl violet stains only the soma or cell body of neuron.

Thank you for pointing out our misunderstanding. We corrected the explanation in Figure 2D according to your comment.

7. Authors are requested to re-evaluate the statistics for PSD95 quantification, in figure 2G,

We overlooked the statistical mistake. We redid the statistical analysis and corrected it as shown in Figure 2G.

8. A β staining is not visible in figure S4C, so it is not clear what the authors have quantified.

The Figure S4C showed that any A β plaques were not observed in young-ages of mice. To show it clearly, we added high-exposure images in Figure S4C.

REVIEWERS' COMMENTS:

Reviewer #1 (Remarks to the Author):

I am very impressed with the careful response to my comments.
The MS is excellent

Reviewer #2 (Remarks to the Author):

The authors have adequately addressed the concerns of this reviewer and improved the visibility of image and graphs in figures.

Reviewer #3 (Remarks to the Author):

The revised manuscript looks much better with improved descriptions. Authors have addressed all the previous comments. Though they still fail to establish a clear explanation : how MITOL regulates toxic amyloid- β oligomer generation (which is their main finding), in the latest version of the manuscript they have tried to discuss the issue and indeed their finding might lead to a new avenue for further research. In my opinion, this article can be accepted for publication.